# Financial Distress, Firm Life Cycle, and Corporate Restructuring Decisions: Evidence from Pakistan's Economy

**Minhas Akbar** [1,2], **Ammar Hussain** [3,*], **Marcela Sokolova** [2] and **Tanazza Sabahat** [1]

1    Department of Management Sciences, COMSATS University Islamabad, Sahiwal Campus, Sahiwal 57000, Pakistan; minhasakbar@cuisahiwal.edu.pk (M.A.); tanazza.sabahat@gmail.com (T.S.)
2    Department of Management, Faculty of Informatics and Management, University of Hradec Králové, Rokitanského 62, 500 03 Hradec Králové, Czech Republic; marcela.sokolova@uhk.cz
3    Department of Management Sciences, Government College University Faisalabad, Sahiwal Campus, Sahiwal 57000, Pakistan
*    Correspondence: amhussain572@gmail.com; Tel.: +92-322-7070682

**Abstract:** This study examines the influence of financial distress on corporate restructuring decisions and whether this restructuring varies across the Firm Life Cycle (FLC) stages of Pakistani non-financial listed firms for the 12 years from 2005 to 2016 inclusive. FLC stages and financial distress are measured using the Dickinson model and Altman Z-score, respectively. Corporate restructuring is segregated into equity and debt restructuring. The data are analyzed using a panel logistic regression model. The results reveal that financial distress is negatively associated with corporate debt restructuring decisions and positively associated with corporate equity restructuring decisions. Further analysis shows that new, growing and mature firms have positive associations with equity restructuring decisions and negative associations with debt restructuring decisions, while declining firms prefer debt restructuring. This study has important implications for corporate managers and policy makers.

**Keywords:** financial distress; corporate restructuring; firm life cycle; logistic regression; Pakistan

## 1. Introduction

Norley et al. (2001) define corporate restructuring as follows: "Restructuring is the process of reorganizing the operational, legal, ownership or other structure of a firm with the intention of making more profit and better organized for its present needs." "Corporate Restructuring is the process of making changes in the composition of a firm's one or more business portfolios in order to have a more profitable enterprise" (Business Jargons 2018). The purpose of restructuring is increased profitability and efficiency. There are different restructuring strategies, which include managerial, financial, operational and asset restructuring. However, after the financial crisis of 2007–2009 the focus on financial restructuring activities increased. Berger (2015) found that, during the crisis period, approximately 9000 German corporations faced financial distress, and that more than 50% of these firms used financial restructuring strategies to survive this financial distress. Financial restructuring strategies include a substantial change in the debt and equity financing patterns of a firm. Financial restructuring is a key factor for firms' success and helps them avoid bankruptcy.

Financial stress occurs when a company's liquidation of aggregate assets is lower than its aggregate estimation of debts (Chen et al. 1995). "Financial distress is a condition in which a company or individual cannot generate revenue or income because it is unable to meet or cannot pay its financial obligations. This is generally due to high fixed costs, illiquid assets, or revenues sensitive to economic downturns" (Kenton 2019). Whenever delayed, this circumstance can prompt liquidation or insolvency (Hendel 1996). Companies in distress need to combine a range of financing instruments to find the optimum solution. Restructuring creates a financial basis that enables value-adding investments and escape

from bankruptcy (Berger 2015). Selective financial restructuring creates more leeway for investments and helps companies avoid falling into the liquidity trap (Berger 2015). Therefore, keeping in mind the importance of financial restructuring and distress in the literature, the aim of this study is to examine the role of the FLC and financial distress in determining corporate restructuring policies.

Scholars describe how firms are conceived, established (built up or framed), grow, achieve maturity, decline and sometimes die (Lester et al. 2003; Ahmed et al. 2021). Firm Life Cycle (FLC) theory is founded in organizational science and especially in Resource Based Theory (RBT). This theory was developed by Penrose (1959) and presents a general direction of the development of a firm while contending that a firms' development relies upon its resources and opportunities. In a similar vein, Wernerfelt (1984) proposed that resources are a definitive root for building and maintaining a competitive edge. Helfat (2003) provided a new dimension for RBT. They argued that firms' resources and opportunities are not static, but rather continue to emerge and progress with the passage of time. This gave birth to the dynamic view of resource-based theory, which is the backbone of the FLC concept. The FLC comprises stages that are outcomes of alterations in external and internal factors (e.g., competitive environment, financial resources, managerial ability, etc.); most of these are a result of strategies undertaken by the firm (Dickinson 2011). Different financial activities determine firm life cycle stages, and have vast implications regarding the understanding of firms' performance and organizational competitiveness (Hasan and Habib 2017). Miller and Friesen (1980) proposed a four stage FLC model, which consists of birth and revival stages along with growth and maturity. However, Gort and Klepper (1982) divide the firm life cycle into five unique stages. This was further verified and supported by Dickenson (Dickinson 2011).

Corporate restructuring decisions have vital importance in the life of a firm. A firm needs to remain afloat in the market by generating cash-flow, making profit and meeting expenses; a firms' survival depends on these corporate restructuring strategies (Ahsan et al. 2016; Koh et al. 2015). Modigliani and Miller (1958) presented the capital structure theory, which became the backbone of a number of theories that further explained corporate restructuring. Financial decision-making changes depend on different firm life cycle stages (La Rocca et al. 2011). It is therefore of utmost importance to study how firm life cycle stages influence corporate structures; this will help policy makers draft strategies accordingly (Ahsan et al. 2016). Furthermore, numerous studies document that idiosyncratic volatility (Hasan and Habib 2017), earnings quality (Hussain et al. 2020), corporate risk-taking (Shahzad et al. 2019), organization capital (Hasan 2018), institutional ownership (Wang et al. 2021), debt maturities (Zhang and Xu 2021), bankruptcy (Durana et al. 2021) and corporate financial decisions (Alqahtani et al. 2021) vary across the firm life cycle. Therefore, the authors believe that corporate restructuring may also vary across the life cycle stages of a firm.

This study contributes to the literature in many ways. Firstly, as far as the authors are aware, this is the first study which empirically explores the role of the FLC and financial distress on corporate restructuring policies (debt restructuring vs. equity restructuring) in the context of an emerging economy, i.e., Pakistan. Secondly, examining the variations in corporate restructuring decisions across FLC stages is a fairly new topic in the financial literature. Thirdly, this present research may be viewed in the context of its auxiliary validation of the Dickinson (2011) measure, which, to the best of our knowledge, has remained unexplored in this relationship. Finally, this research is unique in that it is conducted in Pakistan. Pakistan was chosen due to the fact that it has only a 17% private-sector-credit-to-GDP ratio; this percentage is much lower than that of its neighbors India (48.8%) and Bangladesh (47.6%) (Ahmed et al. 2021). This illustrates that corporations in Pakistan have very limited options regarding financial restructuring.

The rest of this study is arranged as follows: Section 2 critically evaluates the literature and develops our hypothesis; Section 3 describes the methods used; results and discussion are presented in Section 4; and finally, Section 5 concludes the study and discusses its implications.

## 2. Literature Review

### 2.1. Financial Distress and Corporate Restructuring

Different restructuring strategies can be used in financial distress, but financial restructuring is more likely to be used because it can help a firm get better position in the market, expand its business through sound financial decisions and improve its financial operations (Koh et al. 2015). Cash generation strategies, e.g., asset divestment and equity issues, are commonly used strategies to alleviate financial distress, pay down borrowings, reduce interest cost and improve cash flow (Slatter 1984; Sudarsanam and Lai 2001). Distressed firms face severe financial issues and are unable to meet their financial obligations on time, which pushes them towards bankruptcy (Koh et al. 2015; Sudarsanam and Lai 2001). Diamond (1989) asserts that, when firms have lesser financial ability to meet their obligations, it becomes difficult for them to get out of debt. Similarly, such firms develop a negative credit history, and according to Ahsan et al. (2016), a firm's debt financing depends on its past records and history.

Fluck et al. (1998) concluded that firms in the initial stages do not have any reputation or experience, making debt financing difficult. Helwege and Liang (1996) and Ahsan et al. (2016) found that when firms become unable to restructure through debt financing, they use equity financing. Sudarsanam and Lai (2001) concluded that distressed companies raise equity funds via share issues more than non-distressed firms because of pressure from creditors concerned with the security of their lending. Another reason can be higher costs associated with debt financing, which keeps distressed firms away from debt and attracted to equity financing.

Further, Renssen (2017) contended that when a firm is at distress level, it has an option to restructure its business procedures. Koh et al. (2015) documented that these distressed firms are usually motivated toward recoveries through restructuring, and for restructuring, the focus is most often on reducing dividends and investments. Generally, when firms are in financial distress, shareholders threaten the lender and force concessions to avoid possible liquidation of the firm (Tan and Luo 2021). Hence, in the case of financial distress, debt negotiation is the optimal solution (Tan and Luo 2021). Moreover, financially distressed firms have higher bankruptcy risk (Akbar et al. 2019), so financial restructuring is the best tactic to protect the firm from financial distress (ElBannan 2021). Furthermore, there has been an increase in financing through the markets, thus, the importance of financial restructuring is going to be correspondingly higher in future distress cases (Berger 2015). Sometimes, financially distressed firms also use earnings management practices to hide their distressed position or convey a positive signal to investors. In this situation, firms may have the option to use debt restructuring practices and equity restructuring practices (Hussain et al. 2022; Hussain and Akbar 2022) for raising their finances at a low cost. Although several empirical studies have been recently conducted on the determinants of a firm's financial restructuring decisions, the answers remain elusive, and mixed outcomes have been reported (Ahsan et al. 2016). Further, Pakistan is an emerging economy, so the domestic debt to GDP ratio is much lower than other emerging (China) and developed economies (US and UK). As per the World Bank database, 'Domestic credit to private sector (% of GDP)' continuously deteriorated from 21.41% to 16.53% from 2010 to 2016 compared to China, where it increased from 127% to 155% in the same time period. This might be mainly because of high interest rates and tough conditions for credit from the lenders' side. Based on the above statistical figures, the following hypothesis can be developed.

**Hypothesis H1 (H1).** *Financially distressed firms use equity restructuring strategies to become a viable corporation.*

### 2.2. Firm Life Cycle and Corporate Restructuring

Corporate restructuring is an important financial decision for a firm, which keeps on changing depending upon its FLC stage (Ahsan et al. 2016). Chandler (1962) asserts that a firm's strategies and capital structure vary during different stages of its life. Lifecycle theory

recommends that suitable development and capital strategies shift at various phases of a FLC (Anthony and Ramesh 1992). There are four main corporate restructuring strategies, namely; managerial, operational, asset and financial restructuring (Sudarsanam and Lai 2001). Financial restructuring is a way to reshuffle the capital structure, which fundamentally contains value capital and obligation capital. Financial restructuring is divided into two domains, debt and equity restructuring (Sudarsanam and Lai 2001). Berman and Knight (2009) hold that debt financing is an advantage to the organization. Notably, debt restructuring might be very tempting because government bodies incorporate less interest in organizational income, which is very helpful for companies. Besides, such financing is less hazardous and less expensive than 'equity financing' in terms of the rate of return (Berman and Knight 2009). A firm might need to rebuild its capital if one of the two segments of the capital structure exceeds the other (Bowman et al. 1999).

Firms at the introduction and growth stages have more chances to receive finance through equity than through debt. Myers (1984) posits that firms try to balance the portion of debt in overall capital through different stages of a firm's life cycle. This point of view was further verified by (Holmes and Kent 1991; Michaelas et al. 1999; La Rocca et al. 2011). Firms at earlier stages of their life cycle are less capable of holding bigger debts and liabilities (Diamond 1989). As indicated by Diamond (1989), firms at earlier stages have lower obligation abilities than developed firms since introductory firms don't have past track records, while mature firms do have prior histories (Ahsan et al. 2016). Conversely, it is comparatively easier for firms at the introductory and growth stages to restructure through equity financing. An investor can show tolerance and patience for long-term monetary profits, with the expectation that these new and growing firms will pay off for their investment. However, banks and other debt issuers aren't able to show the same level of pateince. The investor expertly underpins new and growing firms with their monetary assets (Kaplan and Strömberg 2003). In this unique circumstance, Carey et al. (1994) demonstrated that small firms will occasionally issue equity before they go into debt. Usually, firms at the initial levels of the life cycle will use their equity before borrowing or taking debts. They will prefer to use venture capital or retained earnings, while debt financing would be their least preferred option (Helwege and Liang 1996; Kaplan and Strömberg 2003). Fluck et al. (1998) concluded that firms at the initial stages do not have any reputation or experience through which they can lend money through debt financing. A Similar point of view was given by (Diamond 1989). Berger and Udell (1998) contended that financing through debt, because of the higher financing cost assumed by banks to counter the higher likelihood of default, is exorbitant for firms at the introduction and growth stages. This can dissuade small firms from utilizing external financing due to information asymmetry or lack of collateral (Weinberg 1994). These firms don't have enough proof that their cash flows will support debt installments, and small organizations are normally not ready to produce positive cash flows in the introduction and growth stages. Therefore, early stage firms depend on family equity financing, i.e., family capital and bank capital that is dependent on family agreements. Thus, as indicated by (Fluck et al. 1998; Helwege and Liang 1996), earlier stage firms are financed chiefly by insiders and by capital venture funding.

At the maturity stage, firms have the option to access both debt and equity financing. Borrowing through the bank is normally made easier after a firm has accumulated substantial resources that may be collateralized. The utilization of debt in the later stages of the FLC turns out to be especially vital (Berger and Udell 1998). When firms approach the post-maturity stages (shakeout and decline), they often try to rebalance their capital structure. As a firm develops, equity can be substituted for debt (Hamilton and Fox 1998). When firms are at the introduction and growth stages, it is difficult for them to get outside financing. Thus, they tend to gather financing from internal sources and through equity financing (Ahsan et al. 2016; La Rocca et al. 2011). As far as debt financing is concerned, when firms grow, their reputation grows, and it becomes easier for them to get lending from banks or through the public. Therefore, debt financing is higher at later stages of a firm's

life cycle (Ahsan et al. 2016). As a firm experiences its life cycle, developing to a stagfe with less information asymmetry, its financing decisions change, often being informed by better access to debt financing options (Chittenden et al. 1996). In this way, the life-cycle example of firm financing accepts that declining firms will be more likely to use the debt financing option (Lerner et al. 2003).

The above literature leads us to understand that firms in the earlier stages of their life cycle adopt equity restructuring strategies to fulfill financial needs, while debt restructuring strategies are more often used during the decline stage. By lending support to the above stated research findings, we formulate the following hypothesis.

**Hypothesis H2a (H2a).** *Compared to the shake-out stage of the firm life cycle, equity restructuring strategies will be used in the introduction and growth stages of the firm life cycle.*

**Hypothesis H2b (H2b).** *Compared to the shake-out stage of the firm life cycle, mature firms can choose both debt and equity restructuring strategies.*

**Hypothesis H2c (H2c).** *Compared to the shake-out stage of the firm life cycle, debt restructuring strategies will be used in the decline stage of the firm life cycle instead of in earlier stages.*

## 3. Methods

### 3.1. Data

The initial sample of the study consisted of 369 non-financial listed firms in Pakistan. The study included only those firms that had five consecutive years of data, allowing for the measurement of the dependent, independent, and control variables. All other firms were eliminated. Therefore, the final sample of 351 firms for a period of 12 years, spanning from 2005 to 2016, was used. Data were collected from the Balance Sheet Analysis (BSA) provided by the State Bank of Pakistan (SBP). The details of the sample firms are presented in Table 1.

**Table 1.** Industry Wise sample distribution.

| Industries Names | No. of Companies |
|---|---|
| 1. Textiles | 134 |
| (a) Spinning, weaving, finishing of textiles | 119 |
| (b) Made up textile articles | 4 |
| (c) Other textiles | 11 |
| 2. Sugar | 28 |
| 3. Food | 16 |
| 4. Chemical, chemical products and pharmaceuticals | 42 |
| 5. Manufacturing | 28 |
| 6. Minerals products | 9 |
| 7. Cement | 17 |
| 8. Motor vehicles, trailers and auto parts | 16 |
| 9. Fuel and energy | 20 |
| 10. Information, communication and transport services | 9 |
| 11. Coke and refined petroleum products | 10 |
| 12. Paper, paperboard and products | 7 |
| 13. Electrical machinery and apparatus | 7 |
| 14. Other services activities | 8 |
| Total | 351 |

### 3.2. Empirical Models

To examine the impact of financial distress and FLC stages on corporate financial restructuring, we will use the following models:

$$FR_{it} = \alpha_0 + \alpha_1 FD_{it} + \alpha_2 TQ_{it} + \alpha_3 size_{it} + \alpha_4 CF_{it} + \varepsilon_i \tag{1}$$

Here, $FR_{it}$ is the financial restructuring strategy (debt and equity restructuring) at current year. $FD_{it}$ represents the financial distress of firm $i$ at time $t$. $TQ_{it}$ is the Tobin's $Q$; $size_{it}$ is the log of total assets of the firm; $CF_{it}$ is the cash flow of the firm; and $\varepsilon_i$ is the error term.

$$FR_{it} = \alpha_0 + \sum_{i=1}^{4} \beta_i FLCS_{i,t} + \alpha_5 TQ_{it} + \alpha_6 size_{it} + \alpha_7 CF_{it} + \varepsilon_i \tag{2}$$

$\sum_{i=1}^{4} \beta_i FLCS_{i,t}$ is a dummy variable used to measure the four stages of the firm life cycle (introduction, growth, maturity, and decline).

### 3.3. Variables Measurement

3.3.1. Firm Life Cycle Stages

We use the (Dickinson 2011) model for the identification of life cycle stages. It presents a dynamic view of the firm life cycle. Dickinson (2011) used data from cash flow statements of the firm to introduce a measurement of firm life cycle stages. She proposes that a firm's cash flow statements show differences in its growth, profitability, and risk. Therefore, firm life cycle stages segregate into introduction, growth, maturity, shakeout, and decline stages. Thus, we use cash from the firm's operating, investing, and financing activities. This classification of firm life cycle stages combines the implications of different research areas of economic literature such as; learning/experience (Spence 1979), entry/exit patterns (Caves 1998), production behavior (Spence 1977, 1979; Wernerfelt 1985), market share (Wernerfelt 1985), and investment (Jovanovic 1982; Spence 1977, 1979; Wernerfelt 1985). In addition, she argues that the cash-flow measure of firm life cycle stages enables us to understand the non-sequential transition of stages that cannot be captured using prior sequential proxies. We classify all of the sample firms into different life cycle stages based on the following cash flow pattern:

(1)    Introduction: if OPCF < 0, INCF < 0 and FCF > 0;
(2)    Growth: if OPCF > 0, INCF < 0 and FCF > 0;
(3)    Mature: if OPCF > 0, INCF < 0 and FCF < 0;
(4)    Decline: if OPCF < 0, INCF > 0 and FCF _ or _ 0; and
(5)    Shake-out: the remaining firm years are classified into the shake-out stage.

where

OPCF = Cash flow from Operations
INCF = Cash flow from investing activities
FCF = Cash flow from financing activities

Consistent with the following studies, (Ahmed et al. 2021; Akbar et al. 2019; Hussain et al. 2020; Wang et al. 2020) considered the shake-out stage of the life cycle as the base stage from which to compare the results of other stages.

3.3.2. Financial Restructuring Strategies (Debt and Equity)

In order to measure the financial restructuring strategies, we created the following dummy variable for financial restructuring (Koh et al. 2015).

$NetDebt_{i,t}$ "Dummy variable that is equal to 1 if Net Debt exceeds 5% of the book value of total asset at year t or t + 1, and zero otherwise."

$NetEquity_{i,t}$ "Dummy variable that is equal to 1 if Net Equity exceeds 5% of the book value of total asset at year t or t + 1, and zero otherwise."

3.3.3. Financial Distress

For measuring the firms' financial distress level, this study used the Altman (1968) Z-score model. Altman (1968) Z-score can help in measuring the financial health of a firm by measuring different balance-sheet values and the firm's income. Altman (1968) developed a model by using five explanatory variables called the Altman Z-score. A Z-score value lower than 1.8 indicates that the firm is in distress. A Z-score value between 1.81 and 2.99 indicates that the firm is in the "caution" zone. A Z-score over 3.0 indicates that the firm is in the safe zone. Z score is measured by the following formula;

Z = 1.2(working capital/total assets) + 1.4(retained earnings/total assets) + 3.3(earnings before interest and taxes/total assets) + 0.6(market value of equity/book value of total debt) + 0.9(sales/total assets).

To assess the distress, many models have been developed after (Altman 1968), but in the last few years, Altman Z-score has proven to be the most accurate for evaluating the financial position of firms (Almamy et al. 2015; Bhandari and Iyer 2013; Chouhan et al. 2014; Mizan and Hossain 2014).

3.3.4. Control Variables

The study uses different control variables that are the major determinants of corporate restructuring (i.e., Tobin's *Q*, firm size, and cash flows *CF*). The Tobin' *Q* (*TQ*) is the ratio equal to the market value of a firm divided by its assets, consistent with (Nazir et al. 2022). Tobin's *Q* is used as a proxy of market pricing. Literature proved the positive relation between the Tobin's *Q* ratio and the performance of a firm (Singhal et al. 2016). *TQ* is calculated as follows:

$$TQ_{it} = \frac{(Market\,capitalization + Total\,Asset - common/ordinary)}{Total\,Asset\,at\,year\,t}$$

Firm size is one of the most significant factors in determining efficiency. High returns and low cost is considered for large-scale production. Therefore, there has been a tendency to grow faster, as regards the size of the firm's industrial units, which will organize mass production and bulk income in diverse markets. Size is measured through the logarithm of total assets. Furthermore, the cash flow of a firm is the net amount of cash and cash equivalents being transferred into and out of a firm. At the most fundamental scale, a firm's capacity to create value for shareholders is decided by way of its potential to generate effective cash flow, or more specifically, to maximize long-term cash flow. Cash flow as a control variable is measured as follows:

$$CF_{it} = \frac{Net\,CasH\,flow\,from\,operations}{Total\,Assets\,at\,year\,t}$$

*3.4. Econometric Approach*

To examine the effects of financial distress and CLC stages on financial restructuring, we employed the panel logistic regression technique. Logistic regression is a method that tries to model the unilateral dependence of variables in which the examined dependent variables are binary, ordinal, or categorical, and the explanatory variables can be of any type (Gregova et al. 2020).

**4. Results and Discussion**

Table 2 reports the descriptive statistic of the variables. Results reveal that a mean ER (equity restructuring) value of 0.92 is greater than a DR (debt restructuring) value of 0.58. Statistics also highlight that the majority of the sampled firms belong to the maturity stage of FLC, and the lowest number of firms belong to the decline phase of FLC. Further, financial distress has an average value of 5.54.

**Table 2.** Descriptive Statistics.

| Variable | Mean | Std. Dev. | Min | Max |
|---|---|---|---|---|
| DR | 0.5799 | 0.4936 | 0.0000 | 1.0000 |
| ER | 0.9163 | 0.2770 | 0.0000 | 1.0000 |
| Intro | 0.1779 | 0.3825 | 0.0000 | 1.0000 |
| Growth | 0.1813 | 0.3853 | 0.0000 | 1.0000 |
| Mature | 0.4464 | 0.4972 | 0.0000 | 1.0000 |
| Shakeout | 0.1300 | 0.3364 | 0.0000 | 1.0000 |
| Decline | 0.0643 | 0.2453 | 0.0000 | 1.0000 |
| FD | 5.5412 | 9.6689 | −12.3883 | 234.0211 |
| CF | 0.0487 | 0.5073 | −29.4069 | 3.6772 |
| Size | 6.5626 | 0.6940 | 4.1088 | 8.7975 |
| TQ | 7.1845 | 14.1744 | −1.8526 | 386.9090 |

Note: DR = debt restructuring; ER = equity restructuring; Intro, Growth, Mature, Shakeout and Decline represents the FLC stages; FD = financial distress; CF = cash flow from operations; Size = firm size; and TQ = Tobin Q.

Table 3 presents the correlation matrix for the variables. It reveals that all the corelative values are fairly less than + or −0.70, as per recommended threshold presented by (Kervin 1995), beyond which, a multicollinearity issue could persist.

**Table 3.** Correlation Matrix.

| | | | | | | | | | | |
|---|---|---|---|---|---|---|---|---|---|---|
| DR | 1.00 | | | | | | | | | |
| ER | −0.11 | 1.00 | | | | | | | | |
| Intro | 0.02 | −0.07 | 1.00 | | | | | | | |
| Growth | 0.05 | 0.08 | −0.22 | 1.00 | | | | | | |
| Mature | −0.06 | 0.11 | −0.42 | −0.42 | 1.00 | | | | | |
| Shakeout | −0.01 | −0.02 | −0.18 | −0.18 | −0.35 | 1.00 | | | | |
| Decline | 0.02 | −0.22 | −0.12 | −0.12 | −0.24 | −0.10 | 1.00 | | | |
| FD | −0.03 | 0.18 | −0.15 | −0.06 | 0.18 | 0.05 | −0.10 | 1.00 | | |
| CF | 0.02 | 0.09 | −0.15 | 0.02 | 0.14 | −0.01 | −0.06 | 0.08 | 1.00 | |
| Size | 0.17 | 0.16 | −0.08 | 0.07 | 0.07 | −0.01 | −0.12 | 0.12 | 0.08 | 1.00 |
| TQ | −0.02 | 0.11 | −0.12 | −0.05 | 0.14 | 0.04 | −0.07 | 0.99 | 0.07 | 0.10 | 1.00 |

Table 4 reports the regression estimations where the financial-distress stage and four other different FLC stages are separately regressed on debt restructuring with a set of control variables. Column 1 reports that financial distress is negatively associated with debt restructuring ($p < 0.01$). It implies that, in case of financial distress, firms are not motivated to move towards debt restructuring strategies. This implies that in Pakistan, when a firm lies under financial distress, lenders are reluctant to give loans, hence, firms are motivated to adopt equity restructuring strategies to recover their existing positions. These results are also supported by Koh et al. (2015), who documented that financially distressed firms are usually motivated toward recoveries through a restructuring that is more often focused on the reduction in dividends and investments.

**Table 4.** Regression Results Dependent Variable: Debt Restructuring.

| | (1) | (2) |
|---|---|---|
| | DR | DR |
| FD | −0.521 *** | |
| | (−6.69) | |
| Intro | | −0.603 *** |
| | | (−3.05) |
| Growth | | −0.435 ** |
| | | (−2.18) |
| Mature | | −0.411 ** |
| | | (−2.39) |
| Decline | | 0.648 ** |
| | | (2.58) |
| CF | 4.237 *** | 2.645 *** |
| | (7.08) | (4.54) |
| Size | 10.48 *** | 9.597 *** |
| | (20.13) | (22.32) |
| TQ | 0.355 *** | 0.0487 *** |
| | (6.98) | (4.12) |
| Prob. value | 0.000 | 0.000 |
| LR $Chi^2$ | 1102.91(4) | 1139.35(7) |
| N | 2383 | 3089 |

*** and ** indicates 1% and 5% significant level respectively.

Further, column 2 documents that with coefficient values of (−0.603, −0.435, and −0.411), the association of the introduction, growth, and maturity stages of FLC is negative and significant with debt restructuring, respectively. Negative values show that there is an inverse relationship between FLC stages (except decline) and debt restructuring strategies.

In contrast, results report that, with a coefficient value of (0.648,) the response to the decline stage is positive and significant ($p < 0.05$) for debt restructuring, which implies that firms use a debt restructuring strategy during the decline stage of FLC, supporting (H2c). These results are also in line with the argument of La Rocca et al. (2011), who conducted a study on how firms make capital restructuring during their FLC, and found that, at the later stage, firm reputation also grows, and it becomes easier to get debt from banks or through the public.

Table 5 reports the regression estimations. Column 1 presents the estimations where financial distress is regressed on equity restructuring with a list of control variables. In contrast to Table 4 estimations, outcomes reveal that the response of financial distress to equity restructuring is positive and significant. This implies that, in the case of financial distress, firms are usually motivated to utilize equity restructuring instead of debt restructuring strategies. In column 2, we regress different FLC stages on equity restructuring practices along with some control variables. The results reveal that the introduction and decline stages of FLC have an insignificant association with equity restructuring practices. In contrast, the response of the growth and maturity phases towards equity restructuring is positive and significant ($p < 0.05$). In light of the above results, we can say that the equity financial restructuring strategy is used in the growth and maturity stages of FLC, as the coefficients of the introduction and decline phases are insignificant. These results are consistent with the following studies (Akbar et al. 2019; Ahsan et al. 2016; Diamond 1989; Berger and Udell 1998; La Rocca et al. 2011). These researchers emphasize that firms in the shakeout and decline stages prefer debt financing because they have built reputations and histories, so it becomes easier for them to secure debt (La Rocca et al. 2011). Further, as mentioned in $H_{2b}$, firms at the maturity stage have the luxury of choosing between either of the strategies (debt and equity). Diamond (1989) assessed the phenomena of 'Reputation Acquisition in Debt Markets' and concluded that firms at the earlier stages of their life cycle are less capable of holding bigger debts and liabilities, and are less obligated to their own development, since firms in the introduction and growth stages don't have past track records, while mature firms do have previous histories. Helwege and Liang (1996) found that firms in the earlier stages are financed mainly through insiders and capital venture funding.

**Table 5.** Regression Results Dependent Variable: Equity Restructuring.

|  | (1) | (2) |
|---|---|---|
|  | ER | ER |
| Intro |  | 0.292 |
|  |  | (0.71) |
| Growth |  | 1.209 ** |
|  |  | (2.48) |
| Mature |  | 0.862 ** |
|  |  | (2.24) |
| Decline |  | −0.424 |
|  |  | (−0.98) |
| FD | 2.131 *** |  |
|  | (7.35) |  |
| CF | 0.130 | 0.949 * |
|  | (0.23) | (1.81) |
| size | 5.575 *** | 4.359 *** |
|  | (5.08) | (5.76) |
| TQ | −0.768 *** | 0.332 *** |
|  | (−4.50) | (3.82) |
| Prob. Value | 0.000 | 0.000 |
| LR $Chi^2$ | 230.47(4) | 98.78(7) |
| $N$ | 495 | 593 |

***, **, and * indicates 1%, 5% & 10% significant level respectively.

## 5. Conclusions

Examining the influence of financial distress and FLC stages on different firm-related financial decisions has always been a key interest of researchers. In this study, we embarked on an effort to investigate the impact of these two important strands of literature on corporate financial restructuring strategies. For this purpose, we collected the data of Pakistani non-financial listed firms spanning from 2005 to 2016, and employed a panel Logistic model as an econometric approach. Results reveal that, compared to the shake-out stage of the FLC, the responses of the introduction, growth, and maturity stages of FLC towards debt restructuring strategies is negative, while the response of the decline stage is positive. In contrast, in the growth and maturity stages, firms are positively engaged in equity restructuring, while in the introduction and decline stages, their responses are insignificant towards equity restructuring. Moreover, estimated results show that the response of financial distress is negative (positive) to debt (equity) restructuring strategies.

Our research has also some important implications, which are equally beneficial for policy makers, investors, and managers. Firm managers should frame their financial restructuring strategies according to the respective FLC stage of that firm because at each phase of the FLC, priorities and conditions towards financial restructuring vary significantly. This implies that policy makers should make effective and reliable long-term financial decisions that are consistent with the life cycle stage of the firm. The present research has also some limitations. Firstly, the current study is conducted on only a developing economy by utilizing data from the twelve years spanning from 2005 to 2016. Thus, a generalization of the results should be made with caution. Secondly, this study did not consider the industry-wise effects. Thus, an industry-wise analysis in the future would help to illuminate this relationship more deeply.

**Author Contributions:** Conceptualization, M.A.; methodology and software, A.H.; validation, T.S.; writing—review and editing, M.S. All authors have read and agreed to the published version of the manuscript.

**Funding:** This article is supported by the Excellence 2022 project, run at the Faculty of Informatics and Management, University of Hradec Kralove, Czech Republic.

**Data Availability Statement:** Data will available on the request to corresponding author.

**Conflicts of Interest:** The authors declare no conflict of interest.

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
