# Peer review of "Financial Distress, Firm Life Cycle, and Corporate Restructuring Decisions: Evidence from Pakistan’s Economy"

_economies, doi:10.3390/economies10070175_

Round 1

Reviewer 1 Report

Dear author(s),

Good paper. Just some comments and questions:

  1. Norley, Swanson, and Marshall, (2001) (line 18): please take out ","
  2. remain afloat in market by making profits (line 73): I think you may want to also include cashflow. Cash is the king.

  3. capital structuring theory (line 75): it should be "capital structure theory"

  4. the first study which empirically 92 explores the role of FLC and financial distress on corporates restructuring policies (debt 93 restructuring vs equity restructuring) (line 93-94): - corporate?; - you cited Koh et at (Journal of Corporate Finance, vol 33, 2015, 19-33), whose sample is US firms from 1995 to 2013; could you elaborate more on your statement on "the first study"?

  5.  Growth: if OPCF > 0, INCF < 0 and FCF > 0; (line 261): What is the justification for OPCF > 0? Usually when the firm at the growth, the OPCF < 0; thus, it is covered by FCF > 0.

    Thank you. 

Author Response

Hi Dear Reviewer,

Please see the attached response sheet in word format.

Thanks in Advance

Reviewer 2 Report

The paper submitted for consideration to Economies examines the relationship between financial distress and corporate restructuring decisions, and the moderating role of the firm life cycle stages in Pakistani listed firms. The author/s find/s (hereinafter in plural) a negative relationship with debt decisions and a positive one with equity decisions.  The life cycle stage affects significantly this decision. The paper reviews a big amount of literature, uses a reliable dataset and provides results coherent with the theoretical assumptions. On the whole, the paper could be an interesting contribution provided that some improvements were introduced. The suggestions are set below:

Major issues

  1. Introduction: The introduction is too long and should go more to the point. Not until the fifth paragraph (out of six) is the aim of the study introduced. This section of the paper should present more concisely the motivation and main contributions of the research. Similarly, the first paragraph of section 2.1 is obvious and could be dropped out.
  2. Hypotheses: In the same vein, the last couple of sentence before H1 are also obvious (there is no doubt that financially distressed firms have higher bankruptcy risks), so the authors should look for more solid foundations for this hypothesis. This fact raises a concern about the novelty of H1 since equity seems to be the only possibility for financially distressed firms.
  3. The references to bank debt in the third paragraph of section 2.2 are confusing and must be better presented in the framework of the life cycle theory.
  4. There are some arguable assertions that require a better explanation. For instance, the idea that debt financing can be more expensive and drive the firms towards equity (third paragraph of section 2.1) is inconsistent with some references and ideas that are introduced later on. The second paragraph of section 2.2 states that in “real world scenarios, firms …”. I am afraid that most of the prior literature studies the firms in real world.
  5. Dataset: the sample consists of 351 non-financial firms but more information about the representativeness of the sample should be added. Are the initial 369 firms the whole population of non-financial listed firms in Pakistan? Have the firms been selected according to any criteria? How much of the market capitalization does the sample account for?
  6. According to Table 2, 58 percent of the sample firms have done a debt based restructuration and 91 percent of the firms have done an equity based restructuration. In this case, can it be said that almost all the firms (91%) have taken an equity restructuring decision? These numbers are too high and cast doubts on the identification strategy of financial restructuring decisions. The identification is based on Koh et al., (2015) but these authors acknowledge that they are not mutually excluding recovery strategies.
  7. The section on discussion can be shortened and merged with the section on results.

Minor issues

  1. The reference format must be reviewed: there are some references with omitted information (e.g., Dickinson, 2011), the journal name sometimes is in italics, sometimes with abbreviations, etc.

To sum up, this paper can be an interesting research provided that some critical (non-fatal) points were amended. Specially, the authors must introduce more convincing rationales for the empirical methodology. I wish the authors good luck with this research, and hope them to find my comments helpful to improve the paper.

Author Response

(The authors gave the same response as above.)

Round 2

Reviewer 2 Report

The authors have addressed most of my concerns, so I deem the paper can be accepted.